# ALS-Related Mutant SOD1 Aggregates Interfere with Mitophagy by Sequestering the Autophagy Receptor Optineurin

**DOI:** 10.3390/ijms21207525

**Published:** 2020-10-13

**Authors:** Yeong Jin Tak, Ju-Hwang Park, Hyangshuk Rhim, Seongman Kang

**Affiliations:** 1Division of Life Sciences, Korea University, Seoul 02841, Korea; xkr1124@korea.ac.kr (Y.J.T.); pjhch7@hanmail.net (J.-H.P.); 2Department of Biomedicine and Health Sciences, College of Medicine, The Catholic University of Korea, Seoul 137-701, Korea; hrhim@catholic.ac.kr; 3Department of Medical Life Sciences, College of Medicine, The Catholic University of Korea, Seoul 137-701, Korea

**Keywords:** amyotrophic lateral sclerosis (ALS), superoxide dismutase 1 (SOD1), optineurin (OPTN), mitophagy

## Abstract

Amyotrophic lateral sclerosis (ALS) is a neurodegenerative disease characterized by the progressive demise of motor neurons. One of the causes of familial ALS is the mutation of the gene encoding superoxide dismutase 1 (SOD1), which leads to abnormal protein aggregates. How SOD1 aggregation drives ALS is still poorly understood. Recently, ALS pathogenesis has been functionally implicated in mitophagy, specifically the clearance of damaged mitochondria. Here, to understand this mechanism, we investigated the relationship between the mitophagy receptor optineurin and SOD1 aggregates. We found that mutant SOD1 (mSOD1) proteins associate with and then sequester optineurin, which is required to form the mitophagosomes, to aggregates in N2a cells. Optineurin recruitment into mSOD1 aggregates resulted in a reduced mitophagy flux. Furthermore, we observed that an exogenous augmentation of optineurin alleviated the cellular cytotoxicity induced by mSOD1. Taken together, these studies demonstrate that ALS-linked mutations in SOD1 interfere with the mitophagy process through optineurin sequestration, suggesting that the accumulation of damaged mitochondria may play a crucial role in the pathophysiological mechanisms contributing to ALS.

## 1. Introduction

Amyotrophic lateral sclerosis (ALS) is a late-onset neurodegenerative disease characterized by the progressive death of both lower and upper motor neurons in the brain and spinal cord [1,2]. Continuous deterioration of the muscular-related nervous system causes defective swallowing, speaking, and breathing and muscle weakness in ALS. Except in the case of mild progression, ALS typically leads to death within 3–5 years after disease onset [3,4]. Multiple aspects of ALS pathogenesis have been suggested, including oxidative stress, endoplasmic reticulum stress, excitotoxicity, proteasome inhibition, mitochondrial dysfunction, altered axonal transport, and protein aggregation, but the precise mechanism of ALS has not been elucidated [5,6,7]. Most cases of ALS occur without a detectable hereditary factor, but mutations in numerous genes have been linked to several cases of familial ALS (fALS) [8]. Approximately 90% of cases are sporadic (sALS), but 10% of ALS cases are familial. Thus far, mutations in more than 20 different genes have been identified as contributing to ALS [9].

ALS cases onset by mutations in the gene encoding superoxide dismutase 1 (SOD1) account for about 20% of familial ALS cases [10]. To date, more than 150 mutation types of SOD1 have been linked to fALS, with varying rates of aggregation propensity, such as A4V, G85R, and G93A [11]. ALS-linked SOD1 mutations have been reported to affect post translational modifications, such as the dimer affinity, insertion of zinc and copper ions, and intramolecular disulfide bond formation to varying degrees, resulting in aggressive protein aggregation [12,13,14,15]. Mutant SOD1 aggregates lead to co-aggregation with essential cytoplasmic components, impairment of the proteasome function through clogging, and damage of cytoplasmic organelles by aggregation onto the cytoplasmic surface, in addition to abnormal activation of nonneuronal cell types, including astrocytes and microglia [16,17,18,19]. 

Abnormal insoluble protein aggregation is a hallmark of several neurodegenerative diseases, including Alzheimer’s disease (AD), Parkinson’s disease (PD), Huntington’s disease (HD), and ALS [20,21]. There is growing evidence that protein aggregation, such as extracellular accumulation of amyloid-β (Aβ) and intraneuronal hyperphosphorylated tau in AD, synaptic protein α-synuclein deposition (referred to as Lewy bodies) in the neuronal cells in PD, and expanded polyglutamine (polyQ) accumulation in HD and ALS involving the inclusion of SOD1, TDP-43, and FUS, induce neuronal loss through a toxic gain of function [22,23,24,25,26]. However, it remains unclear how these insoluble protein aggregates cause disease.

Optineurin (OPTN) is a highly conserved, 64 kDa hexameric protein consisting of 577 amino acids (aa) that interacts with numerous proteins related to inflammation, protein trafficking by vesicles, and signal transduction, such as the NF-κB pathway [27,28]. OPTN consists of multiple coiled-coil domains, ubiquitin-binding domain (UBD), leucine-zipper kinase, and LC3-binding domain [29]. Previous studies have shown that specific mutations in OPTN are associated with neurodegenerative diseases like glaucoma and ALS. Exon 5 deletion, Q398 nonsense, and E478G missense mutations of OPTN have been reported in patients with familial ALS [30]. Interestingly, OPTN is implicated in several forms of selective autophagy, including aggrephagy and mitophagy [31,32,33]. Mitophagy is vital for neuronal homeostasis, by which damaged mitochondria are selectively sequestered and eliminated through the autophagic process [34]. It has been demonstrated that ALS-associated E478G ubiquitin binding deficient mutations significantly decrease the recruitment of OPTN to damaged mitochondria, resulting in impaired mitophagy [35]. The report suggests that ALS-associated mutant OPTN, through a loss of function, could work as a causative factor for mitochondrial dysfunction, leading to ALS pathogenesis.

Here, we tried to elucidate the pathological mechanism of the ALS-linked relationship between mutant SOD1 aggregates and OPTN. OPTN proteins were well co-aggregated with and sequestered by cytosolic mutant SOD1 aggregates in the neuronal cells. These aggregated OPTNs were further characterized by measuring protein mobility and degradation tendency. Furthermore, we found that the sequestered OPTN could not be recruited to the mitophagosome digesting damaged mitochondria, resulting in an impaired mitophagy process in the presence of mutant SOD1 aggregates. These results provide novel insights into ALS pathogenesis. 

## 2. Results

### 2.1. Optineurin Associates with Mutant SOD1 Proteins, A4V and G93A

It was demonstrated that mutations in the gene encoding OPTN were found in neuronal cells of some familial and sporadic ALS patients, in which OPTN proteins formed protein inclusions, as shown in ALS-linked mutant proteins such as SOD1 and TDP-43 [30]. First, we performed co-immunoprecipitation (Co-IP) assays to investigate the possible association between the common SOD1 mutants (A4V and G93A) and OPTN. In N2a cells transfected with wild type SOD1, A4V, and G93A, respectively, we undertook Co-IP with antibodies against OPTN and then, observed strong interactions of endogenous OPTN with the mutant SOD1 proteins, A4V and G93A, but not with wild type (Figure 1A).

Furthermore, co-localization of endogenous OPTN with mutant SOD1 in N2a cells was investigated using immunofluorescence staining assays and confocal microscopy techniques. As shown in Figure 1B, OPTN proteins were co-localized with A4V and G93A mutant proteins, respectively. On the contrary, OPTN did not co-localize with wild type SOD1. Taken together, these results indicate that OPTN interacts with the ALS-linked mutant SOD1 proteins, A4V and G93A. 

### 2.2. Insoluble Optineurin Significantly Increases in N2a Cells Containing Mutant SOD1 Aggregates

ALS-linked mutant SOD1 proteins have been found to form detergent-insoluble aggregates, and these aggregates are known to induce toxicity through aberrant interactions with normal cellular proteins or organelles [17,36,37]. The co-precipitation and co-localization of OPTN with mutant SOD1 aggregates hints that OPTN could be detected in insoluble fractions, and that the function of OPTN may be disrupted through co-aggregation. To confirm this, we transiently transfected N2a cells with OPTN-GFP and Cerulean-tagged wild type or mutant SOD1 (A4V, G93A) for 48 h. In the presence of wild type SOD1, the OPTN-GFP proteins were distributed in the cytoplasm in the dispersed or puncta form, while in the presence of mutant SOD1, the OPTN-GFP proteins co-aggregated with mutant SOD1 (A4V, G93A), occupying a specific region within cells (Figure 2A, left panel). To determine the ratio of aggregated OPTN in a given population of cells, we counted cells that contained OPTN co-localized with SOD1. This co-aggregation was rarely detected in cells expressing wild type SOD1, compared to cells expressing mutant SOD1 G93A or A4V (Figure 2B).

Furthermore, using fractionation assays of cell lysates, we separated soluble proteins from insoluble proteins. A significant amount of OPTN was found in the insoluble fraction of mutant SOD1 expressing cells, whereas in wild type SOD1 expressing cells, OPTN was found less in the insoluble fraction (Figure 2C,D).

We treated cells with Carbonyl cyanide m-chlorophenyl hydrazone (CCCP), a mitochondrial oxidative phosphorylation uncoupler that can artificially damage mitochondria. Even under these conditions, OPTN still co-aggregated with mutant SOD1 (Figure 2A,B), and the amount of insoluble fraction was almost unchanged compared to cells without CCCP (Figure 2C,D).

Lastly, we carried out fluorescence loss in photobleaching (FLIP) assays in cells expressing OPTN-GFP with wild type or mutant SOD1, to confirm whether the OPTN protein loses its mobility and activity due to OPTN-mutant SOD1 co-aggregation. In the FLIP assay, a specific region of interest (ROI) was continuously bleached, and the loss of fluorescence in the surrounding area was measured over time. The measurement of GFP-tagged OPTN showed that in cells expressing wild type SOD1, time-dependent loss in the photobleached regions was similar to that of unbleached areas, suggesting that OPTN moves at a high frequency. However, when mutant SOD1 aggregates were formed in the cells, the fluorescence intensity of ROI within the aggregates significantly decreased, whereas the unbleached region did not alter, indicating that the mobility of OPTN was diminished (Figure 2E). Collectively, these results demonstrate that OPTNs are enriched in mutant SOD1 aggregates and sequestered through co-aggregation. 

### 2.3. Mutant SOD1 Aggregates Sequester OPTN and Affect the Mitophagy Process

The clearance of damaged mitochondria is essential for cellular homeostasis, since defective mitochondria produce excessive reactive oxygen species (ROS) and induce senescence. Impaired mitochondrial quality control could lead to several neurodegenerative diseases, including ALS [38,39,40]. We hypothesized that mutant SOD1 aggregates sequester OPTN that otherwise would be used for the mitophagy process, and that disrupted OPTN activity causes an accumulation of damaged mitochondria, leading to cell death. 

First, as shown in Figure 3, we induced damage to mitochondria using CCCP. Damaged mitochondria would be captured by autophagosome-containing OPTN, and then, removed by autolysosome. We measured the levels of OPTN in N2a cells after treatment with CCCP at different times and concentrations. The relative amounts of OPTN proteins decreased, and the ratio of LC3II/LCI increased, by 180 min after CCCP treatment (Figure 3A). These phenomena worked in a concentration-dependent manner (Figure 3B). When 20 mM of NH_4_Cl as an inhibitor of autophagy, which blocks the acidification of lysosomes, was used to treat the cells, the amount of OPTN increased (Figure 3A,B). The ratio of LC3II/LCI was higher in the presence of NH_4_Cl. These results suggest that damaged mitochondria are removed through the mitophagy process by using the mitophagy receptor OPTN, but blocking the mitophagy cascade using a lysosomal inhibitor leads to an accumulation of OPTN.

To investigate the effect of mutant SOD1 or aggregates on OPTN activity, we transfected N2a cells with wild type or mutant SOD1 (A4V, G93A), and the amount of OPTN was determined by the Western blotting method. In the presence of mutant SOD1 A4V or G93A, elevated amounts of OPTN were observed compared to wild type SOD1. When the transfected cells were treated with CCCP, the amount of OPTN was reduced in the presence of wild type SOD1. However, the amount of OPTN did not change in the cells expressing mutant SOD1 in CCCP’s presence (Figure 3C). 

The amount of OPTN mRNA was analyzed to investigate whether the increased OPTN protein level was due to elevated transcripts. We observed that the OPTN transcript level was almost unchanged in all cases, indicating that mutant SOD1 or aggregate does not affect the transcription level of OPTN (Figure 3D). These results show that the increased OPTN is due to OPTN protein stability and not due to the mRNA level. Mutant SOD1 or aggregates physically sequester OPTN, resulting in OPTN accumulation in cells. 

Furthermore, we investigated whether the increased OPTN in the presence of mutant SOD1 is observed in human G93A transgenic (Tg) mice. The mouse that we used was paralyzed in its hind legs after 16 weeks and later showed worsened ALS symptoms, such as spine bends and weight loss [41]. We observed that the amounts of OPTN increased and also accumulated in the extracted spinal cords of 16-week-old ALS Tg mice, compared to non-Tg mice (Figure 3E). Thus, mutant SOD1 aggregates cause the accumulation of OPTN in human G93A transgenic (Tg) mice.

### 2.4. Mutant SOD1 Aggregates Inhibit Mitophagosome Formation and Mitophagy Flux 

To further investigate the mechanism by which SOD1 mutants inhibit the mitophagy process in cells, we analyzed the mitophagy flux and formation ratio of mitophagosomes. When mitochondria are damaged, surrounding mitophagy receptor proteins are recruited to mitochondria and form autophagosome-like mitophagosomes. We used cell imaging techniques to examine the dynamics of the mitophagosome formation process. To stimulate mitophagy, N2a cells expressing exogenous Parkin, the E3 ligase required for OPTN-mediated mitochondrial clearance, and SOD1, were treated with 20 μM CCCP. At 120 min after CCCP treatment, OPTN and LC3, an autophagic marker protein, were enriched in the mitochondria. Interestingly, the number of mitophagosomes significantly decreased in cells expressing mutant SOD1 compared to cells expressing wild type SOD1 (Figure 4A,B). We also observed that OPTN and LC3 were distributed throughout the cytoplasm of the cells containing wild type SOD1, whereas in mutant SOD1 expressing cells, these proteins were kept in a locus of the cytoplasm, indicating that OPTN and LC3 may be recruited to aggregates of SOD1 (Figure 4A). 

Mitophagy is the process by which damaged mitochondria and related receptor proteins are disassembled and removed. We monitored the degradation of TOM40, located in the outer membrane, and TIM23, in the inner membrane protein of mitochondria, to examine whether mitophagy worked efficiently in a time-dependent manner. In cells with mutant SOD1, higher levels of OPTN were observed at 8 h after CCCP treatment. Additionally, increased levels of TOM40, TIM23, and exogenous Parkin were detected at 24 h in cells expressing mutant SOD1 compared to wild type (Figure 4C,D). These results imply that the degradation ratio of each mitochondrial protein and mitophagy receptor decreases when mutant SOD1 is expressed. Taken together, mutant SOD1 aggregates inhibit mitophagosome formation and the mitophagy process.

### 2.5. Over-Expression of OPTN Reduces Cytotoxicity Induced by Mutant SOD1

Mitochondrial dysfunction and morphological alteration have been found within the motor neurons of the spinal cords of ALS Tg mice. Mutant SOD1 has been reported to form insoluble aggregates in mitochondria at the surface of the outer membrane, which leads to mitochondrial damage and excessive ROS production [42,43,44]. Furthermore, impaired clearance of mitochondria due to defective mitophagy has been found in many cases of ALS. Thus, we hypothesized that SOD1 aggregates could cause mitochondrial damage and interfere with mitophagy through inhibition of OPTN, resulting in the accumulation of oxidative stress that contributes to ALS. To attenuate the disruption of mitochondrial dynamics, we over-expressed OPTN in cells and investigated whether exogenous OPTN reduces cell death and ROS production. We used propidium iodide (PI) staining assays to detect the cytotoxicity. The tests showed that PI-positive cells occur more in cells expressing mutant SOD1 than cells with wild type SOD1 after CCCP treatment. However, over-expression of OPTN reduced cell toxicity in cells with all types of SOD1 (Figure 5A). We also determined the ratio of cell death using marker proteins, specifically cleaved caspase-3 and poly (ADP-ribose) polymerase (PARP), and found that the amount of both proteins decreased when OPTN was over-expressed (Figure 5B). 

Next, we measured the ROS level generated by mitochondria using dichlorofluorescein (DCF-DA), which is used to detect intracellular oxidative stress. The DCF-DA signal was higher in mutant SOD1 than wild type SOD1-expressing cells after treatment with CCCP, indicating that mutant SOD1 can generate abnormal ROS production. These results imply that elevation in ROS levels is due to the disruption of OPTN-mediated mitophagy to clear damaged mitochondria caused by mutant SOD1 aggregates. However, in OPTN co-expressing cells, the fluorescence signal significantly decreased compared to the control (Figure 5C). Together, these findings demonstrate that OPTN can ameliorate the cytotoxicity and excessive ROS generation induced by mutant SOD1 aggregates. 

## 3. Discussion

In ALS, progressive accumulation of misfolded proteins such as TDP-43, C9orf72, and SOD1 disrupts cellular homeostasis, resulting in neuronal cell death [1,26]. However, although these protein aggregates are a hallmark of the disease, the precise mechanism underlying neurodegeneration pathology is mostly unknown. In this paper, we describe the deleterious roles of SOD1 aggregates in inducing dysfunctional mitophagy through inhibition of OPTN function. 

Previous studies have shown that OPTN co-localizes with the major causative proteins related to neurodegenerative diseases such as HD, AD, and ALS, and OPTN has been considered to have critical roles in the pathogenic mechanisms of these conditions [30,45]. An ALS-related G93C mutant was reported to bind to the C-terminal coiled-coil domain of OPTN in a ubiquitin-independent manner [32]. We also showed that G93A and A4V mutant forms of SOD1 associate with OPTN (Figure 1). Thus, there may be a common mechanism by which OPTN can associate only with mutant forms, but not with wild type, SOD1. However, future studies will be required to determine which specific motif of mutant SOD1 is involved in its interaction with OPTN.

We previously speculated that the mutant SOD1 aggregates inhibit the function of cellular proteins by direct interaction, as well as by positioning and movement of cellular proteins through sequestrating them into aggregates. This study confirmed that the co-localization and association of mutant SOD1 and OPTN lead to a loss of OPTN function. We detected that the cytoplasmic OPTN punctations were recruited into SOD1 aggregates, and formed protein inclusions in N2a cells expressing mutant SOD1. The sequestrated OPTNs were prone to entering an insoluble state. We tested the reduced protein mobility in cells with SOD1 aggregates (Figure 2). 

The level of OPTN was increased in cells expressing mutant SOD1 (Figure 3). The half-life of OPTN was lengthened while the amount of OPTN transcript remained almost unaffected, suggesting that mutant SOD1 contributes to the post translational regulation of OPTN. Furthermore, we investigated the level of OPTN in G93A transgenic mice at p120, which showed the morphology and symptoms of ALS, including neuronal cytoskeletal pathology [41,46]. We confirmed the presence of accumulated OPTN in the spinal cords extracted from the Tg mice, implying dysregulation of OPTN in this ALS mice model (Figure 3).

Previous studies have shown that the accumulation of dysfunctional mitochondria is considered as a crucial factor for neurodegenerative diseases, including ALS [47]. The impaired clearance of damaged mitochondria via mitophagy has been thought to be associated with the PINK/Parkin and OPTN/TBK1 pathways in these diseases. This study found mutant SOD1 interferes with OPTN-mediated mitophagosome formation and mitophagy flux. CCCP disrupts mitochondrial membrane potential and promotes mitophagy, which induces recruitment of OPTN to mitochondria. However, we observed that the translocation of co-localized OPTN–LC3 complexes to mitochondria decreased at 180 min after CCCP treatment in cells with mutant SOD1 (Figure 4A,B). Previous reports have demonstrated that ALS-associated mutations in TBK1-E696K and OPTN, such as E478G and S177, block efficient mitophagy [35], suggesting that mutant SOD1 aggregates also have a similar detrimental effect on functional OPTN. Furthermore, we analyzed mitophagy flux by monitoring mitochondrial protein degradation in Parkin-mediated mitophagy in cells expressing WT or mutant SOD1, respectively. Surprisingly, we identified that the degradation rates of the mitochondrial membrane proteins TOM40 and TIM23, as well as the mitophagy receptors OPTN and Parkin, were reduced after CCCP treatment (Figure 4C,D). This result indicates that mutant SOD1, which tends to sequester other proteins by building large aggregates, may also affect functional proteins involved in selective autophagy. 

Mitochondria are essential organelles for metabolism and molecular signaling, and mitochondrial impairment is a crucial feature of neurodegenerative diseases, including AD and ALS [48,49,50]. Especially during aging, dysfunctional mitochondria that produce more ROS abnormally accumulate, leading to oxidative stress and cellular damage. Preceding research has reported that mutant SOD1 forms aggregates in the mitochondrial outer membrane and interacts with membrane proteins such as Bcl-2, resulting in an overproduction of ROS [37]. We speculate here a novel reason for the increased ROS in the ALS phenotype. The mitophagy process typically eliminates the damaged mitochondria induced by CCCP treatment. However, the sequestered OPTN proteins caused by SOD1 aggregates are increased in cells expressing mutant SOD1, inhibiting the translocation of OPTN to mitochondria, and leading to the accumulation of more ROS-generating dysfunctional mitochondria. However, when augmentation of OPTN through over-expression in cells is introduced, the cellular toxicity leading to cell death and the levels of intracellular ROS are also alleviated (Figure 5). Therefore, the relationship between OPTN and ALS-linked mutant SOD1 is crucial for ROS homeostasis through mitophagy.

For this study, we used the N2A cell line, mouse neuroblastoma, and SOD1-G93A mice and control mice. Although there have been a number of reports that have used rodent cell lines in order to understand ALS pathogenesis [51,52], OPTN experiments using human-based cell models will provide more clues about the translation of findings to ALS patients.

In conclusion, we propose a mechanism for the dysregulation of OPTN-mediated mitophagy by SOD1 aggregates (Figure 6). In ALS, aggregated mutant SOD1 binds to OPTN and sequesters it by forming large aggregates. Mutant SOD1 can exacerbate the mitophagy process by blocking the translocation of OPTN to dysfunctional mitochondria and interfering with its organelles’ clearance. Decreased turnover of mitochondria via mutant SOD1 could be a critical factor for the accumulation of defective mitochondria, a common feature in ALS. This study suggests that modulation of mitophagy by regulating its interaction between mutant SOD1 and OPTN may be a potential therapeutic approach for ALS. 

## 4. Materials and Methods 

### 4.1. Cell Culture and Transfection 

The mouse neuroblastoma cell line N2a was maintained in Dulbecco’s Modified Eagle Medium (DMEM), supplemented with 10% fetal bovine serum, streptomycin (100 ug/mL), and penicillin (100 U/mL), at 37 °C in a CO_2_ incubator. Transfections were performed with polyethyleneimine (PEI) (Sigma-Aldrich, St Louis, MO, USA), as previously described [53]. DNA was combined with PEI at a ratio of 1:3 and incubated in serum-free DMEM for 30 min. After incubation, DNA-reagent complexes were applied to the cells. The media were replaced with complete media 6 h later. Cells were incubated for an additional 24–48 h at 37 °C in a CO_2_ incubator.

### 4.2. Plasmid Constructions 

SOD1-FLAG, SOD1-GFP, SOD1-Cerulean, and Myc-Parkin-HA constructs have been described previously [54,55,56]. OPTN cDNA fragments were amplified by polymerase chain reaction (PCR), using pfu polymerase and first-strand human cDNAs as templates. OPTN-GFP were generated by inserting amplified OPTN fragments into pEGFP-C2 plasmids at the EcoRI and BamHI restriction sites. OPTN-HA were generated by inserting amplified OPTN fragments into pcDNA-HA-4T-1 plasmids at the EcoRI and XhoI restriction sites, respectively. pDsRed-Mito backbone (Clontech, CA, USA) was used to detect mitochondria. Plasmids encoding GFP-tagged LC3 were gifted from Chungho Kim (Korea University, Seoul, Korea). LC3-GFP were generated by inserting amplified LC3 fragments into pEGFP-C1 plasmids at the HindIII and EcoRI restriction sites. 

### 4.3. Reagents and Antibodies 

The following antibodies were used in the immunoblotting, co-immunoprecipitation, and immunofluorescence assays: HA monoclonal Ab (mAb), GAPDH polyclonal Ab (pAb), TOM40 mAb, and TIM23 mAb (Santa Cruz, CA, USA); OPTN pAb (Abcam, MA, USA); β-actin mAb and Flag mAb (Sigma-Aldrich, MO, USA); and Caspase-3 mAb and PARP Ab (Cell Signaling Technology, MA, USA). Propidium Iodide (PI), Carbonyl cyanide *m*-chlorophenyl hydrazine (CCCP), Cycloheximide (CHX), and 2′,7′-Dichlorofluorescin diacetate (H_2_DCFDA) were obtained from Sigma-Aldrich.

### 4.4. Immunoblotting Assay 

Transfected cells were lysed for 30 min at 4 °C in RIPA buffer (0.15 M NaCl, 0.5% sodium-deoxycholate, 0.1% SDS, 0.05 M Tris-HCl (pH 8.0), 1% NP-40) supplemented with a protease inhibitor. Spinal cords taken from 16-week-old ALS Tg-mice were lysed for 1 h at 4 °C in NETN buffer (100 mM NaCl, 1 mM EDTA, 20 mM Tris-HCl (pH8.0), 0.5% NP-40) containing protease inhibitor. Cell or tissue lysates were quantified with the Bradford assay (Bio-Rad, Hercules, CA, USA). Then, the samples underwent SDS–polyacrylamide gel electrophoresis (SDS–PAGE) and were transferred to nitrocellulose membranes. The membranes were incubated overnight with the indicated primary antibodies at 4 °C. Following incubation with the appropriate secondary antibody for 1 h at room temperature, the proteins were visualized with an enhanced chemiluminescence (ECL) immunoblotting system, as described by the manufacturer (GE Healthcare Life Sciences, Chicago, IL, USA) [53].

### 4.5. Co-Immunoprecipitation (Co-IP) 

Transiently transfected N2a cells were harvested and lysed on ice for 1 h in 1% NP-40 lysis buffer containing protease inhibitor. Cell lysates were centrifuged at 13,000 rpm for 30 min at 4 °C, and the protein extracts were incubated with the indicated antibodies overnight at 4 °C. The immunocomplexes were collected by centrifugation after incubating with Protein G Sepharose beads (GE Healthcare Life Sciences) [57]. The bead-bound proteins were separated by 12% SDS–PAGE and immunoblotted with indicated Abs.

### 4.6. Animals 

All animal experiments were done in compliance with the Institutional Animal Care and Use committee at Korea University (KUIACUC-2019-0014, approval date: 2 January 2019). The transgenic B6SJL-Tg [SOD1-G93A] 1 Gur/J (SOD1-G93A), and B6SJLF1/J control mice used in this study were purchased from the Jackson Laboratory. SOD1-G93A mice were maintained by breeding male hemizygous SOD1-G93A mice with female B6SJLF1/J hybrids. The genotyping of SOD1-G93A mice was performed by PCR, as previously described [41].

### 4.7. Cell Viability Assay

Transfected cells were fixed with 4% formaldehyde in phosphate-buffered saline (PBS) for 5 min at RT. After washing three times with PBS, the fixed cells were incubated with RNase A (100 ug/mL) for 20 min and incubated with PI (250 ng/mL) for 15 min at 37 °C in a CO_2_ incubator. After washing three times with PBS, stained cells were counted using an LSM-700 confocal microscope (Carl Zeiss).

### 4.8. DCF-DA Assay 

Transiently transfected N2a cells were plated on 6-well plates and incubated in the presence of CCCP 10 μM for 12 h. Cells were washed three times with PBS, and then, incubated with 20 μM H_2_DCFDA for 30 min at 37 °C in a dark CO_2_ incubator. After washing three times with PBS, the cells were harvested by trypsinization. Then, suspended cells were washed using a centrifuge at 2500 rpm for 5 min and examined under flow cytometry using the 488 nm laser for excitation and detected at 535 nm. 

### 4.9. Soluble/Insoluble Fraction 

Cells were harvested in 1X TEN (10 mM Tris, 1 mM EDTA, and 100 mM NaCl) mixed with an equal volume of 2X extraction buffer 1 (10 mM Tris, 1 mM EDTA, 100 mM NaCl, 1% NP-40, and protease inhibitor mixture), and then, we sonicated the lysate three times for 1 min on ice between each two-second pulse. After sonication, the cell lysate was centrifuged at 100,000× *g* for 5 min at 4 °C in a Beckman TLR100.3 ultracentrifuge, to separate the detergent-insoluble pellet (P1) from the supernatant (S1). The P1 was resuspended in 1× extraction buffer 2 (10 mM Tris, 1 mM EDTA, 100 mM NaCl, 0.5% NP-40, and protease inhibitor mixture) and sonicated under the same conditions. The re-suspended pellet was then centrifuged for 5 min at 100,000× *g* in an ultracentrifuge to separate the second pellet (P2) from the supernatant. P2 was re-suspended in buffer 3 (10 mM Tris, 1 mM EDTA, 100 mM NaCl, 0.5% NP-40, 0.25% SDS, 0.5% deoxycholic acid, and protease inhibitor mixture) by sonication. S1 represented the soluble fraction, and re-suspended P2 represented the insoluble fraction [17,58]. Protein concentration was measured by the Bradford assay.

### 4.10. FLIP 

FLIP experiments were performed as described previously [55]. Cells were grown in a 35 mm glass-bottomed dish (SPL Life Science Co., Ltd. (Pocheon-si, Korea)). GFP-tagged OPTN and Flag-tagged SOD1 plasmids were transfected into cells. Forty-eight hours after transfection, photobleaching was performed by a 405 nm laser at 50–100% output for ten iterations in live cells. Fluorescent images were captured using an LSM-700 confocal microscope (Carl Zeiss, Oberkochen, Germany).

### 4.11. Immunofluorescence Assay (IFA)

Transfected N2a cells were fixed in 4% formaldehyde for 15 min, rinsed three times with PBS, and quenched for 10 min with 50 mM NH_4_Cl solution. Subsequently, cells were permeabilized for 5 min in PBS containing 0.1% Triton X-100, and incubated in a 2% bovine serum albumin (BSA) containing blocking buffer for 1 h at RT. After rinsing with PBS, cells were incubated for 2 h at RT with OPTN-Ab (1:200) diluted in a blocking buffer. Cells were washed three times in PBS, then incubated in Alexa Fluor 405 (Life Technologies, Thermo Fischer Scientific, Carlsbad, CA, USA), then diluted 1:400 in a blocking buffer for 1 h at room temperature. After washing three times with PBS, processed samples were mounted onto glass slides using fluorescent mounting medium (Vectashield, USA) [17,57]. Fluorescent images were captured using an LSM-700 confocal microscope (Carl Zeiss).

### 4.12. qRT-PCR 

Total RNA was extracted from the transfected cells with the easy-Blue Total RNA Extraction Kit (Intron, Seongnam, Korea), and then, reverse-transcribed into first-strand cDNA using the RevertAid First Strand cDNA Synthesis Kit (Thermo Fischer Scientific, Waltham, CA, USA). A quantitative PCR was performed using the following primers: OPTN forward (5′-TTC AAA GAG AAA TCA GAA AAG CCA-3′) and reverse (5′-CTC CTC CAA GGC TCT GGG A-3′) primer [59]; GAPDH forward (5′-AAC TTT GGC ATT GTG GAA GG-3′) and reverse (5′-ACA CAT TGG GGG TAG GAA CA-3′) [60]. Real-time quantitative PCR was performed with LightCycler^®^ 480 SYBR Green I Master (Roche, Basel, Switzerland) using the LightCycler 480 II system (Roche, Basel, Switzerland) [61].

### 4.13. Statistical Analysis 

All data were presented as the mean ± SEM (standard error of the mean) of three independent experiments. Differences between the various experimental groups were calculated using the Student’s two-tailed *t*-test and one-way analysis of variance (ANOVA) test. P-values of <0.05 were considered statistically significant.

## Figures and Tables

**Figure 1 ijms-21-07525-f001:**
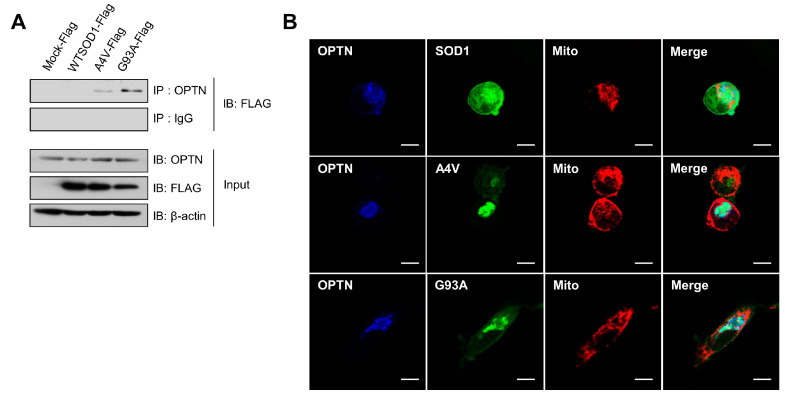
Optineurin associates with the mutant SOD1 proteins, A4V and G93A. (**A**) Immunoprecipitation assay for SOD1 and endogenous OPTN in N2a cells expressing SOD1-FLAG, A4V-FLAG, G93A-FLAG, or FLAG empty vectors. At 48 h after transfection, each sample was immunoprecipitated with OPTN antibodies and immunoblotted with FLAG antibodies. (**B**) N2a cells expressing DsRed-mito, which detects mitochondria, together with SOD1-GFP, A4V-GFP, or G93A-GFP, were immunostained with OPTN antibodies at 48 h after transfection. Scale bar = 10 μM.

**Figure 2 ijms-21-07525-f002:**
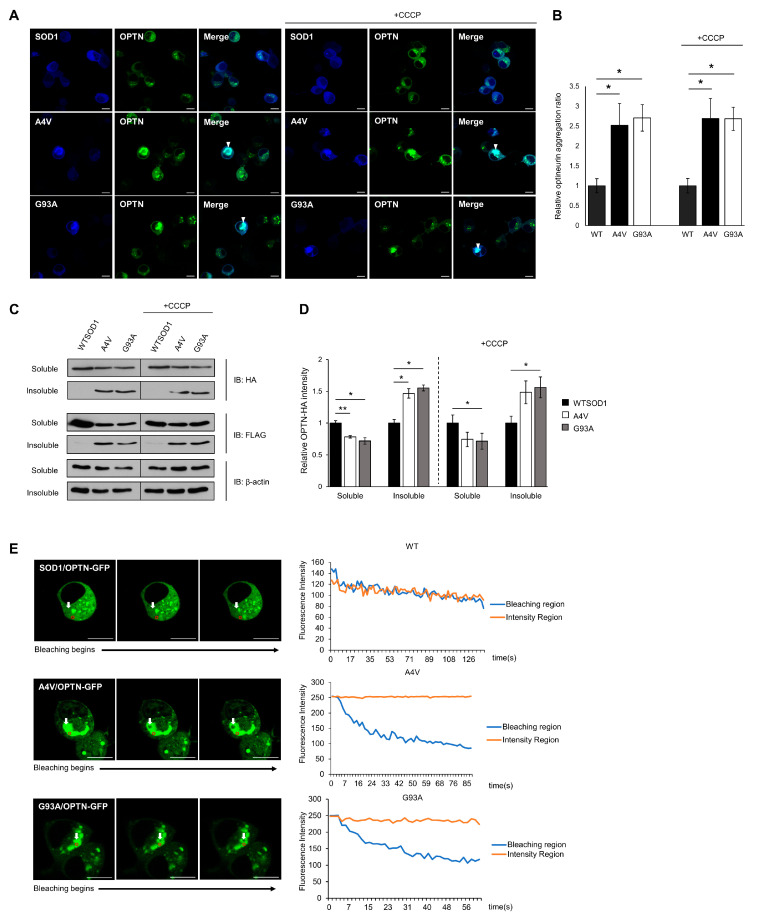
Insoluble optineurin significantly increases in cells with mutant SOD1 aggregates. (**A**,**B**) Counting the number of intracellular OPTNs enriched with SOD1 aggregates in the presence or absence of CCCP treatment (20 μM, 2 h). N2a cells were transiently transfected with OPTN-GFP and Cerulean tagged wild type or mutant SOD1 (A4V, G93A) for 48 h. Arrows indicate OPTN co-aggregated with SOD1. *n* = 250–300 cells per group. Scale bar = 10 μM. (**C**,**D**) Quantitation of soluble and insoluble OPTN in cells transiently transfected with HA-OPTN and FLAG-tagged SOD1, A4V or G93A. At 48 h after transfection, the lysates separated into a soluble and insoluble fraction. Each sample was immunoblotted with anti-HA, anti-FLAG, and anti-β-actin antibody. The results were normalized to β-actin. The results reflect three independent experiments. * *p* < 0.05. ** *p* < 0.01. (**E**) For FLIP assay, N2a cells were transiently transfected with OPTN-GFP together with FLAG-SOD1, FLAG-A4V, or FLAG-G93A. At 48 h after transfection, each cell was bleached, and the intensity of OPTN-GFP was measured. The white arrow indicates regions where bleaching was performed, and the fluorescence in the red circle was measured. Scale bar = 10 μM.

**Figure 3 ijms-21-07525-f003:**
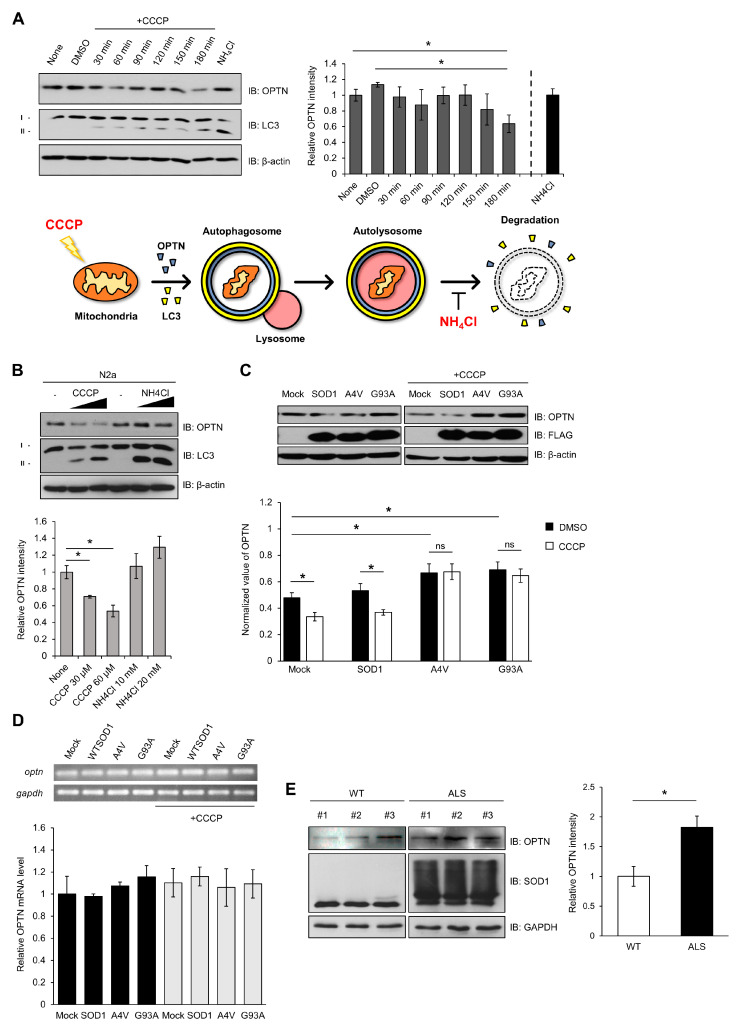
Mutant SOD1 aggregates lead to accumulation of optineurin, required for mitophagy. (**A**) N2a cells were lysed and immunoblotted using anti-OPTN, anti-LC3, and anti-β-actin after being treated with CCCP (30 μM) in a time-dependent manner, or NH_4_Cl (10 mM, 3 h). The results were normalized to β-actin. (**B**) Quantitation of the endogenous OPTN levels in N2a cells following CCCP or NH_4_Cl treatment for 3 h in a different concentration. (**C**) Quantitation of the endogenous OPTN levels in N2a cells expressing FLAG-SOD1, FLAG-A4V, FLAG-G93A, and FLAG. At 48 h after transfection, each sample was immunoblotted against specific antibodies in the absence or presence of CCCP treatment (20 μM, 12 h). (**D**) The levels of OPTN mRNA from the N2a cells expressing FLAG-SOD1, FLAG-A4V, FLAG-G93A, and FLAG were measured using quantitative PCR. CCCP treatment was applied (20 μM, 12 h) to the cells at 48 h post transfection. The results were normalized to GAPDH. (**E**) Quantitation of the OPTN level from the spinal cords of 16-week-old SOD1-G93A mice and control mice. The results were normalized to GAPDH. *n* = 3 per group. * *p* < 0.05.

**Figure 4 ijms-21-07525-f004:**
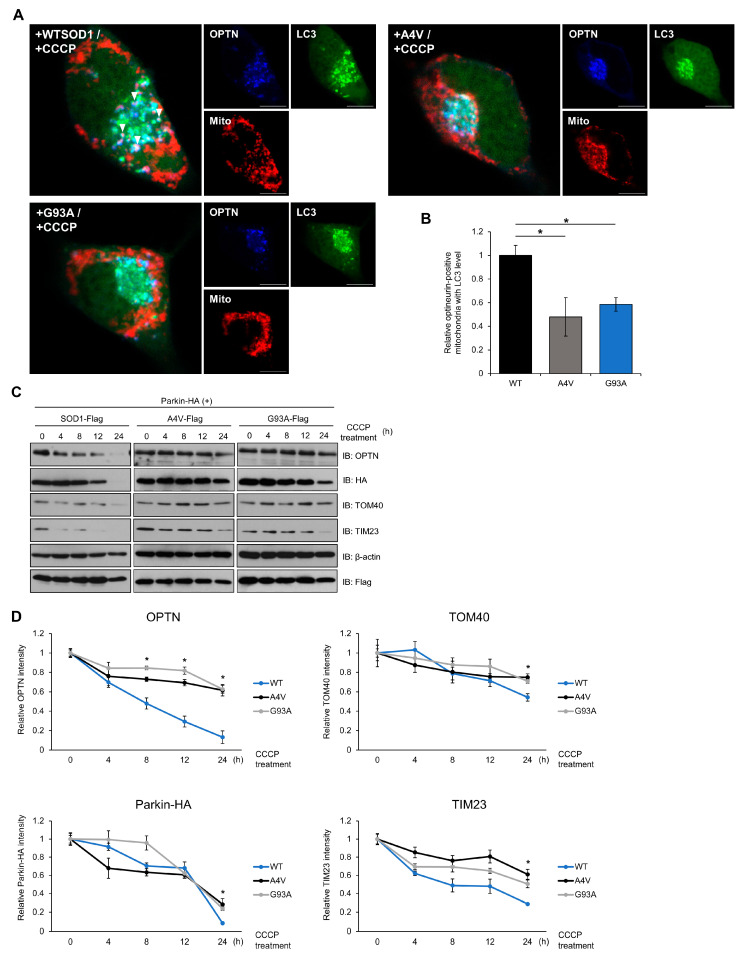
Mutant SOD1 aggregates inhibit mitophagosome formation and mitophagy flux. (**A**,**B**) N2a cells were transiently transfected with LC3-GFP, Myc-Parkin-HA, DsRed-mito, and FLAG-tagged SOD1 (WT, A4V or G93A) for 48 h. Cells were immunostained with anti-OPTN antibodies after CCCP treatment (20 μM, 3 h). Arrows indicate mitochondria surrounded by mitophagosomes. The number of OPTN-positive mitochondria with LC3 was measured by the counting method. *n* = 21–30 cells per group. Scale bar = 10 μM. (**C**,**D**) Mitochondrial protein degradation detected by Western blotting. Treatment with CCCP (20 μM) to induce mitophagy was performed with N2a cells expressing Myc-Parkin-HA and FLAG-SOD1, FLAG-A4V, or FLAG-G93A for various time periods. The lysates were immunoblotted with antibodies against TOM40, TIM23, OPTN, FLAG, HA, and β-actin. The results were normalized to β-actin. * *p* < 0.05.

**Figure 5 ijms-21-07525-f005:**
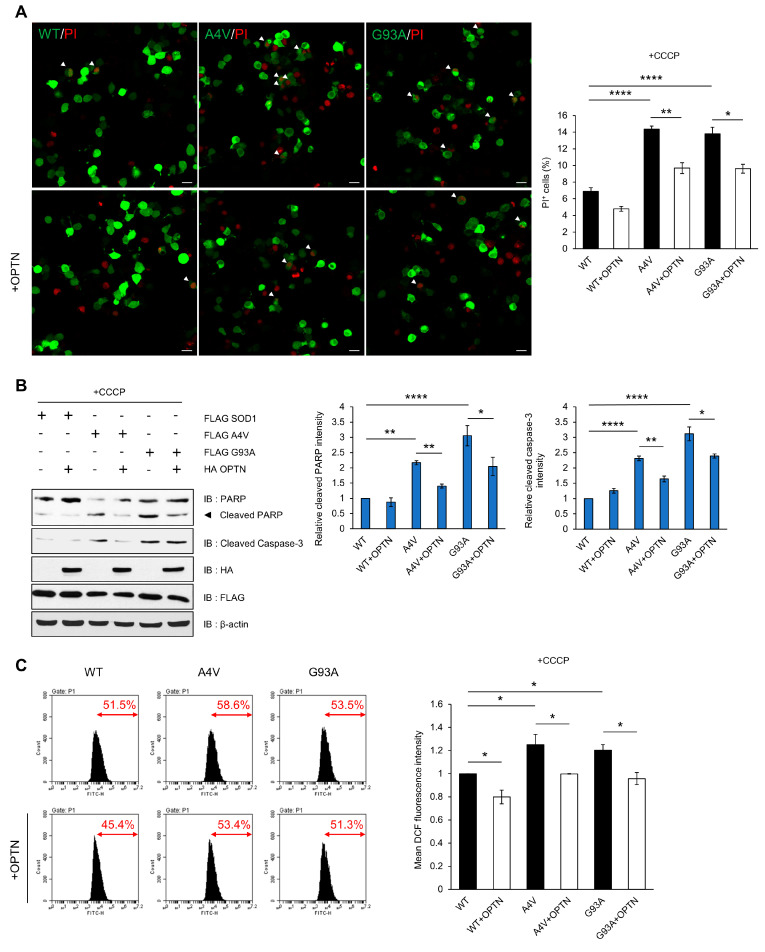
Over-expression of OPTN reduces cytotoxicity induced by mutant SOD1. N2a cells were transiently transfected with FLAG-tagged wild type SOD1 or mutant SOD1 (A4V and G93A) for 48 h. (**A**) Applied post transfection, the PI stained nuclei identifies dead cells. The cell death ratio was measured as the percentage of SOD1-GFP transfected cells exhibiting PI+ staining (white arrows). Scale bar = 20 μM. (**B**) For the validation of cell death, cell lysates were analyzed by immunoblot with antibodies to PARP and caspase-3. (**C**) Intracellular ROS production was measured by DCF fluorescence and analyzed by flow cytometry after CCCP treatment (10 uM, 12 h). The data are means ± SEM from three independent experiments. * *p* < 0.05; ** *p* < 0.01; **** *p* < 0.0001.

**Figure 6 ijms-21-07525-f006:**
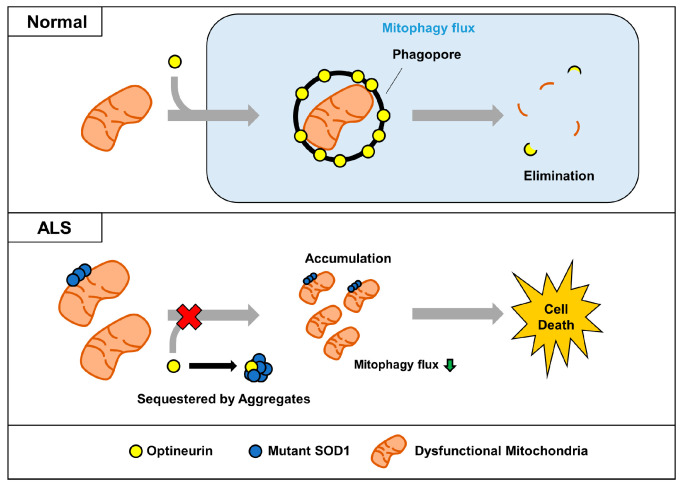
Proposed model for impairment of OPTN-mediated mitophagy by mutant SOD1 aggregates. In normal cells, dysfunctional mitochondria can be captured by the mitophagy receptor OPTN, and removed together. However, sequestered OPTN by SOD1 aggregates cannot be translocated to dysfunctional mitochondria, and this leads to the abnormal accumulation seen in ALS.

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
