# Peer review of "ALS-Related Mutant SOD1 Aggregates Interfere with Mitophagy by Sequestering the Autophagy Receptor Optineurin"

_ijms, 2020, doi:10.3390/ijms21207525_

Round 1

Reviewer 1 Report

The manuscript goal is focused on the inetercation between optinuerin and mutant SOD and the effect on cells (decrease mitophagy). This is an important finding for decipher ALS molecular pathology at least in the familiar form with mutations in SOD. Several experiments are done with two different mutants of SOD and wild type, and also using CCCP as mitophagy inducer. The paper is interesting and the results sound reasonably. 

The major weaknesses is the model used for the experiments: the N2A cell line. This is a a fast-growing mouse neuroblastoma cell line and thus results have low translational impact. The use of human neuroblastoma cell line SH-SY5Y or other human based cell models should be use at least in the first experiments to provide some clues about translation to the patients.

Author Response

Thank you for your consideration of our manuscript entitled ALS-related mutant SOD1 aggregates interfere with mitophagy by sequestering autophagy receptor optineurin” (Manuscript No.: ijms-932833). We greatly appreciate the suggestions of the referees. The co-authors and I have carefully considered the suggestions by them; we agree with most of these points and have revised the manuscript accordingly. In addition, we have improved our manuscript through the MDPI English editing service. We believe that the revised manuscript is substantially stronger than the original and hope that it is now suitable for publication.

Reviewer #1

The manuscript goal is focused on the inetercation between optinuerin and mutant SOD and the effect on cells (decrease mitophagy). This is an important finding for decipher ALS molecular pathology at least in the familiar form with mutations in SOD. Several experiments are done with two different mutants of SOD and wild type, and also using CCCP as mitophagy inducer. The paper is interesting and the results sound reasonably. 

The major weaknesses is the model used for the experiments: the N2A cell line. This is a a fast-growing mouse neuroblastoma cell line and thus results have low translational impact. The use of human neuroblastoma cell line SH-SY5Y or other human based cell models should be use at least in the first experiments to provide some clues about translation to the patients.

Author’s response:

  • As you suggested, we have improved our manuscript through the MDPI English editing service and attach the English-editing certificate document.
  • In this study, we have used the N2A cell line, mouse neuroblastoma, and SOD1-G93A mice and control mice. We used the mouse neuroblastoma: 1) to further confirm our results in SOD1-G93A mice after the cell line experiments, 2) to use the induced pluripotent stem (iPS) cells from fibroblasts of SOD1-G93A mice that we previously made [Park JH, Park HS, Hong S, Kang S. Motor neurons derived from ALS-related mouse iPS cells recapitulate pathological features of ALS. Exp Mol Med. 2016 Dec 9;48(12):e276. doi: 10.1038/emm.2016.113. PMID: 27932790; PMCID: PMC5192071].

We absolutely agree with the reviewer’s suggestion that the use of human neuroblastoma cell line SH-SY5Y or other human based cell models provides some clues about translation to the patients, although there have been reports that used rodent cell lines in order to understand the ALS pathogenesis. So, for the next study, we are going to investigate more human based cell models and patient’s sample to get insight into translation to the patients.

To let readers to know informational limitation of experiments to use the mouse neuroblastoma, we have added a paragraph in the discussion section (page 12 line 331-334).

Reviewer 2 Report

Yeong Jin Tak et al. investigated the link between SOD1 mutation and OPTN. They found that mutant SOD1 disrupts mitophagy through the recruitment of optineurin to SOD1 aggregates. In general the work is interesting and performed well. However, it requires several corrections:

  1. In abstract/text is stated that mutation in SOD1 is a major known cause of familial ALS, this is incorrect, C9orf72 mutation is more common, please correct
  2. Authors used co-IP to assess physical interaction of SOD1 and OPTN, co-IP does not allow to conclude about interaction, please rewrite
  3. Statistics: Figure 3 A B, Figure 5 A B C, ANOVA or suitable non-parametric test will be more appropriate than t-test
  4. Figure 3 B – poor quality blot, particularly for beta-actin

Author Response

 Thank you for your consideration of our manuscript entitled ALS-related mutant SOD1 aggregates interfere with mitophagy by sequestering autophagy receptor optineurin” (Manuscript No.: ijms-932833). We greatly appreciate the suggestions of the referees. The co-authors and I have carefully considered the suggestions by them; we agree with most of these points and have revised the manuscript accordingly. In addition, we have improved our manuscript through the MDPI English editing service and attach the English-editing certificate document. We believe that the revised manuscript is substantially stronger than the original and hope that it is now suitable for publication.

Reviewer #2

Yeong Jin Tak et al. investigated the link between SOD1 mutation and OPTN. They found that mutant SOD1 disrupts mitophagy through the recruitment of optineurin to SOD1 aggregates. In general the work is interesting and performed well. However, it requires several corrections:

  1. In abstract/text is stated that mutation in SOD1 is a major known cause of familial ALS, this is incorrect, C9orf72 mutation is more common, please correct

Author’s response:

As the reviewer suggested, we have corrected the sentence ‘A major known cause of familial ALS is the mutation of the gene encoding superoxide dismutase 1 (SOD1) that leads to abnormal protein aggregates.to ‘One of the causes of familial ALS is the mutation of the gene encoding superoxide dismutase 1 (SOD1) that leads to abnormal protein aggregates.in line number 12.

  1. Authors used co-IP to assess physical interaction of SOD1 and OPTN, co-IP does not allow to conclude about interaction, please rewrite

Author’s response:

As the reviewer requested, we have rewrote the sentence ‘First, we performed co-immunoprecipitation (Co-IP) assays to investigate whether the common SOD1 mutants (A4V and G93A) physically interact with OPTN. to ‘First, we performed co-immunoprecipitation (Co-IP) assays to investigate the possible association between the common SOD1 mutants (A4V and G93A) and OPTN. in line 82-83.

  1. Statistics: Figure 3 A B, Figure 5 A B C, ANOVA or suitable non-parametric test will be more appropriate than t-test

Author’s response:

As the reviewer suggested, we have analyzed using ANOVA instead of t-test in Figure 3 A B and Figure 5 A B C. Accordingly, we have adjusted the p-value of the graph shown in each figure and stated at the end of the legend paragraph; we have added a new sentence about ANOVA in the ‘Statistical Analysis’ of ‘Materials and Methods’ section.

  1. Figure 3 B – poor quality blot, particularly for beta-actin

Author’s response:

As the reviewer suggested, we have replaced the blot images in Figure 3 B with better quality ones.

*In addition, we have elaborated our manuscript through the MDPI English editing service.

Round 2

Reviewer 1 Report

Although more experiments have not been done, the updates provided for authors improved the quality of the paper and merits to be published

Author Response

Reviewer #1’s comment: Although more experiments have not been done, the updates provided for authors improved the quality of the paper and merits to be published

  • Thank you. We appreciate your acceptance comment of our manuscript.